# ‘All We Have to Decide Is What to Do with the Time That Is Given to Us’ a Photovoice Study on Physical Activity in Nursing Homes [note 1]

**DOI:** 10.3390/ijerph18105481

**Published:** 2021-05-20

**Authors:** Dorothée Altmeier, Ansgar Thiel, Annika Frahsa

**Affiliations:** 1Institute of Sport Science, University of Tuebingen, 72072 Tuebingen, Germany; dorothee.altmeier@uni-tuebingen.de (D.A.); ansgar.thiel@uni-tuebingen.de (A.T.); 2Institute of Social and Preventive Medicine, University of Bern, 3012 Bern, Switzerland

**Keywords:** older adults, Photovoice, nursing homes, physical activity, CBPR, nursing staff

## Abstract

(1) Background: Despite high prevalence of physical inactivity and sedentariness among nursing home residents, research on the influence of environments on this topic remains scarce. This Photovoice study explores how structural and social environments relate to residents’ everyday physical activity (PA). (2) Methods: 27 residents, staff, and significant others conducted Photovoice in eight nursing homes in Germany to document factors facilitating or hindering PA. Photographs were discussed with the participants in eight focus groups and analysed using thematic analysis. (3) Results: 169 photographs (between 8 and 42per home) were categorized into three thematic groups: (1) ‘architectural challenges for PA promotion in nursing homes’; (2) ‘opportunities and limitations of using daily work equipment for PA promotion’; (3) ‘social incentives for PA promotion’. Photographs’ foci in the homes differed considerably between participant groups. Staff primarily chose environmental constructions and aids that they perceived to enable residents’ PA. Residents were more likely to express affections and emotions that would encourage them to be active. (4) Conclusions: PA promotion research in this setting should be sensitive to diverse perceptions of different stakeholder groups and existing power imbalances. Interventions are needed that integrate residents’ needs and train staff on how to consider residents’ perspectives.

## 1. Introduction

Sedentary behaviour represents a substantial health risk among older adults [1]. Several studies have indicated a link between sedentary behaviour (prolonged sitting time) and health risks, and among those an increased risk for all-cause mortality, several chronic conditions, e.g., diabetes, different forms of cancer or cardiovascular diseases [2,3], and a decline in physical functions [4,5]. Particularly older adults living in nursing homes sit for long periods of time [6,7,8,9]. 

At the same time, sedentary behaviour represents a social norm in nursing homes, visible in procedures, structures, and behavioural patterns typical in this setting: residents sit during mealtimes, in most activities, and during conversations with relatives in the cafeteria. Sedentary time encompasses 90% of a nursing home resident’s day on average [10]. Only a very limited number of residents in nursing homes meets the recommendations for physical activity (PA) [11] by the World Health Organization [12]. Furthermore, the individual extent of PA among residents is often related to their cognitive status (such as high levels of ‘wandering’ among people with dementia) [8,13]. Given the importance of PA for maintaining health and preventing the progression of nursing home residents’ care dependency, increasing PA in everyday life is crucial for the reduction in sedentary time. 

While several studies with children and adolescents have already documented that the environmental context shapes children’s [14,15] and adolescents’ everyday lives and is likely to influence PA patterns [16], research on environments conducive to PA among older adults remains scarce. Studies on the impact of the neighbourhood’s physical and social environment on older adults’ PA show that safeguarding environments does not suffice to keep older adults physically active [17,18]. Environmental conditions, such as structures conveying a sense of security, play an important role for older adults’s perception of being able to be physically active. However, ensuring residents’ safety may also lead to sedentariness in nursing homes if there is no concurrent social support for being physically active [8,19]. Safety measures can, therefore, both promote and prevent PA. A further major structural barrier to residents’ activation in nursing homes is the lack of time perceived by nursing home staff [20].

The roles other structural barriers or facilitators play regarding PA and sedentariness among nursing home residents have barely been examined to date. The significance of this research gap lies in the fact that interventions aimed at increasing PA in nursing homes tend to not be sustainable if they are not structurally integrated in the nursing homes [21,22]. Hence, there is a need to better understand how the environment in which nursing home residents spend their time may impact sedentary behaviour and PA [23]. 

Against this background, this study investigates which factors in nursing homes influence the residents’ PA. Inspired by a study by Mahmood and colleagues [17], we decided to use Photovoice to document barriers and facilitators to PA in nursing homes. Photovoice is a participatory action research method that builds upon people’s photographs to explore individuals’ life-worlds and community issues, to promote critical consciousness and reflection, and to foster community change and exchange with decisionmakers toward social action, as first described by Wang and Burris [24,25]. 

Since its introduction, Photovoice has been applied for a variety of populations, places, health issues, and disciplines. For example, adolescents used Photovoice to collect their views on health and wellbeing [26], healthy eating, and active lifestyle barriers [27] or information on sexual health, alcohol, and drugs [28,29]. Some studies have applied the Photovoice method in studies with older adults [30,31,32,33]. Studies also presented Photovoice as an opportunity to portray problems people experience who are physically or cognitively limited [34,35,36]. In these studies, Photovoice was often reported to be an appropriate method for older adults to make a statement about their lives, particularly about occurring limitations or problems with regard to their everyday lives, e.g., to find out how to set up an age-friendly area in a neighbourhood or community [17,28]. There are also Photovoice studies that report on barriers to physical activity for older adults in neighbourhood settings [17,32,37]. Studies identified safety, security, accessibility, comfort, destination diversity, community-based programs, peer support, and intergenerational or volunteer activities as the main promoters for PA in older age [17], reported on the use of Photovoice to developing adequate health messages for older adults and minority populations [37], or identified traffic hazards and falls risks as significant barriers to walking for older adults, and connectivity, hazard-free routes, and simple amenities such as benches and washrooms that might promote walking in such an age group [32]. 

Very few Photovoice studies explicitly took place in the context of nursing homes, and if so, they tended to focus on physical, psychological, and social characteristics that characterize the facility as ‘home’ [38], overall values identified as relevant by residents, such as joy of nature, leisure-time activities, and friendship [39], or on the process of conducting Photovoice and building a photonarrative [40].

Nursing homes, at the same time, appear to be particularly promising, and challenging, for Photovoice, for various reasons. If embedded in a Freirean participatory action research context, Photovoice results may stimulate social action [25], such as change towards physical activity promotion in nursing homes. If done appropriately, Photovoice moves beyond needs assessment and enables community members to become advocates for improvements and wellbeing in the respective context [25]. So far, only a very small number of Photovoice projects with older adults have included a mixed sample that systematically involved perspectives of different stakeholder groups [30]. However, while it is decisive to give voice to the residents themselves, it is also crucial to integrate professionals’ perspectives if you aim to increase critical consciousness, reflective practice, and agency for structural change [24,41]. Within a nursing home setting, daily routines and a specific form of lifestyle tend to be predefined by the institution and executed by staff [42]. Therefore, any physical activity promotion is potentially closely interlinked with institutional structures and staff perspectives on this issue.

Based on these considerations, our study aims to explore factors residents, staff, and significant others (such as relatives, friends, and external service providers) considered important for residents to become physically active and to compare the perceptions of the different stakeholder groups. To address these goals, three research questions were phrased to guide future development of adequate PA-promoting action in nursing homes:(1)What are the factors perceived by residents, staff, and significant others that improve or hinder nursing home residents’ PA?(2)Are there differences among the different stakeholder groups?(3)What are the differences between residents’, staff’s, and significant others’ perceptions?

## 2. Methods

### 2.1. Methodological Approach

Based on Paulo Freire’s theories of problem-posing and participatory education [43], Photovoice enables community members to identify relevant problems by collecting visual evidence of significant realities and interacting with researchers in needs assessment to create critical consciousness and stimulate change [25].

To be able to record environmental conditions in their lifeworld related to a certain behaviour or potential health risk (e.g., diabetes) with high ecological validity, participants use cameras or tablets to identify and document resources, barriers, and facilitators [44]. Afterwards, participants meet as a group to discuss photographs. These group discussions are to provide context and explanation of the photographs to identify common themes and concerns. This process, on the one hand, helps researchers to make sense of data and, on the other hand, can also be useful for participants by creating a critical mindset regarding their environment and its impact on behavioural patterns [44,45]. 

An approach that integrates the different relevant perspectives, at the same time, needs to be sensitive to existing power imbalances between residents, significant others (such as relatives, friends, as well as external service providers), and nursing home staff as well as to the risk of social control and surveillance [46]. These risks might pose challenges both to residents but also to staff within very hierarchical institutions such as nursing homes. Thus, researchers should make sure to consider those risks and their potential impacts when analysing photographs taken (and the ones not taken) and group discussions. 

### 2.2. Study Setting

This Photovoice study took place as part of a larger project, the BaSAlt Study on PA Promotion and Counselling in Nursing Homes (*‘Verhältnisorientierte Bewegungsförderung und individuelle Bewegungsberatung im Setting Altenwohnheim’ ein biopsychosoziales Analyse– und Beratungsprojekt’,* funded by the German Federal Ministry of Health 2019–2022, grant no. ZMVI1-2519FSB114). The final sample consisted of eight nursing homes in the state of Baden-Württemberg in Germany. Nursing homes varied regarding their environmental contexts, responsible bodies, and organization forms, as well as capacities and composition of resident populations. Three nursing homes were located in an urban environment and five in more rural areas. We included a convenience sample of eight nursing homes run by four different non-profit institutions. The participating nursing homes had capacities to care for between 33 and 49 residents. The number of residents requiring the maximum level of care (the equivalent to level 4 in the German healthcare system) varied between 13 and 31 in the participating nursing homes (11/2020).

Ethical approval for the study was granted by the Ethics Committee of the Faculty of Economics and Social Sciences at Eberhard Karls University Tübingen (No. AZ A2.5.4-096_aa). The Ethics Committee granted us an amendment to acknowledge adaptations of the study design (assessment procedure and instruments, inclusion of digital elements in assessments and counselling, required safety measures to minimize the risk of spreading COVID-19).

Personal data were collected and stored in accordance with the European Data Protection Basic Regulation (DSGVO) and in coordination with the data protection officers of the institutions involved. Data were treated confidentially and pseudonymized for the process. All participants provided written informed consent to participate in the study.

### 2.3. Recruitment of Photographers

We recruited six residents, 10 staff members, and eight significant others (four relatives, two physiotherapists, one relative of a deceased resident, one pastor) in eight nursing homes involved in the larger BaSAlt project. Participants had to meet the following criteria: They had to (a) have a connection to the respective nursing home, (b) be able to take digital photographs, and (c) be willing to participate in a 1 to 1.5-h-long focus group discussion. Initially, 27 participants signed up, 1 staff member and 2 residents dropped out of the study due to decreasing interest or acute health issues. The participants’ age ranged from 18 to 91 years, with women (n = 16) outnumbering men (n = 8). This distribution reflects the gender distribution in the nursing homes with regard to residents and staff, as well as to visitors.

The process of participant recruitment was supervised by a member of the research team. Staff were informed at the respective facilities and invited to participate as photographers. Flyers were handed out to recruit participants. In addition, staff members suggested residents and relatives potentially interested in documenting hindering and fostering social and environmental factors for PA in the individual homes. Residents were mostly recruited during breakfast time by one research team member. The majority of significant others were suggested and informed by the nursing homes’ executive staff and recruited by a research team member. Recruitment criteria for all perspectives were that they were familiar with daily life at the specific nursing home and that they could provide an insider perspective on options and barriers as well as that they were physically, cognitively, and verbally able to take photographs with a tablet and discuss their results.

### 2.4. Photovoice Procedure and Analysis

The Photovoice study took place between February and November 2020 to cover all seasons and their potential benefits and specific barriers to PA.

The participants were asked to take photographs of situations, objects, and places during their daily routines documenting when and why residents became physically active as well as barriers that would stop residents from becoming active. 

Participants were allowed to take photographs both inside and outside the nursing home. There was neither a minimum nor a maximum of photographs defined to be taken. Participants were supported in the data collection process and, upon request, assisted in taking a photograph by researchers familiar with the locations. Researchers took fieldnotes during the process.

The first author assisted participants who needed help operating the tablets. For a later understanding of why the photographs had been taken, directly after the respective photo session, the following survey questions had to be completed for each photograph:What title would you give this photograph?Do you consider what you photographed to be a?
Fostering activity?Hindering activity?What do you want to express with the photograph?Decide on the potential of the photograph’s content to foster activity:
The extent to which activity is fostered in percentage.The extent to which activity is hindered in percentage.

With participants who were unable or unwilling to fill out the questionnaire, we conducted an oral interview instead and noted down their answers in the questionnaire as well as taking fieldnotes.

All photographs and the corresponding explanations were compiled in a presentation format for later comparison with the focus group discussion results.

Due to the COVID-19 pandemic, it was impossible to conduct the focus group discussion face-to-face in the nursing homes; hence, we decided to execute the process remotely. We held training sessions at the nursing homes with at least one nursing home representative, either on-site or online via video conference software to guide them on how to inform about the focus group process, motivate participants to work with the sheets, and to moderate the group discussion. In case of questions, a member of the researcher team was available for giving information. 

We conducted at least one focus group in each of the eight nursing homes. All participants of the Photovoice study were invited to join the discussion. Although we aimed to include all relevant groups in the focus group discussions, it was not possible for significant others to participate due to COVID-19-related contact restrictions. Participants could decide either to work on the focus group questions in self-organised discussions within a period of three weeks or in pre-planned, scheduled sessions. If a nursing home chose the first option, sheets with instructions and all the photographs, which had been numbered and reproduced in a 13 × 18 cm^2^ format, were on display at a prominent location in the nursing home for a three-week-long period; otherwise, the focus group discussions were held in team sessions with up to 10 participants or in prescheduled small groups (2–4 persons). 

Three work instructions guided the focus group discussion: (1) evaluate whether what was photographed promotes residents’ PA (indicated by putting a round green sticker next to the photograph) or hinders it (indicated by a round red sticker); (2) discuss the rationale for assigning a photograph to one of the groups; (3) decide about the time of the day (morning, afternoon, evening) during which the photographed situation/factor occur. 

Discussion results were visually recorded on the sheets, as well as documented in field notes during a debriefing session with the responsible nursing home representative.

We conducted reflexive thematic analysis [47,48] to interpret all data (photographs, fieldnotes, documentation from focus group discussions) with multiple steps (anchor quotes used in the results section, from survey responses, interviews, fieldnotes, and documentations, cf. quotes in original language and English translation Appendix A). Researchers then met for critical reflection and to further analyse notes, headings of photographs, and group discussion results into broader themes. The final categorization of themes across all nursing homes is presented in the results.

## 3. Results

In total, 169 photographs were taken, varying between a minimum of seven and a maximum of 42 photographs per home (cf. an overview, Table 1). 

Photographs showed (a) a variety of mobility assistive equipment; (b) group mobilizations; (c) physical therapists or medical clowns (external service providers) at work; (d) children, animals, handicrafts, and plants; (e) doorways, elevators, corridors, stairways, and bathrooms; (f) musical instruments and songbooks.

Noticeably more artefacts were photographed than people. Some photographs were ambiguous, e.g., a defective elevator that could both hinder (residents’) PA or increase (staff’s) PA. 

In the following, we present (1) three main themes derived from the data; (2) differences regarding the photographs taken by residents, staff, and visitors; (3) statements on the perceived relevance of the photographs; (4) insights into what was not photographed.

### 3.1. Main Themes

Photographs and relating explanations were categorized into the following overarching themes. Three main themes were identified: (1) ‘architectural challenges of PA promotion in nursing homes’; (2) ‘opportunities and limitations of daily work equipment for PA promotion’; (3) ‘social incentives for PA promotion’.

#### 3.1.1. Architectural Challenges of PA Promotion in Nursing Homes

The theme ‘architectural challenges of PA promotion in nursing homes’ (cf. examples Figure 1) includes findings relating to the indoor and outdoor designs of the nursing homes. The German Industrial DIN norm requires nursing homes to be barrier-free facilities, providing wheelchair users with sufficient space to manoeuvre. A large number of the photographs by staff members and significant others document this status quo. However, the photographs do not show whether the DIN standard-compliant facilities in fact allow residents to move around. Particularly in the photographs taken by the staff members, the focus was rather on building defects such as thresholds inherent in the construction of the building that limit the residents’ PA.

Numerous examples for PA enhancing infrastructure norm standards by staff members featured long and wide corridors, which according to the WHO, are particularly well suited to promote PA. 


*Long stretches without obstacles promote physical activity, as they can be used by residents with wheelchairs and walkers regardless of weather conditions.*
[staff member 1]

At the same time, participating staff reported that corridors tend to be closed off by doors, which was considered a significant barrier to PA.


*Closed doors are impedimental because the desire for physical activity is halted. ‘you shall not pass’*
[staff member 2]

Obstacles in the corridors, such as cleaning trolleys or laundry boxes, were further limiting factors for PA documented by the participating staff and significant others. These objects often turned the corridor into an obstacle course. 

Limited space in public areas, such as spatial restrictions in the corridors but also in the dining areas, were also considered as hindering for PA by a relative.


*Not enough space is restrictive for PA, there is no space to turn around with a wheelchair or a rollator.*
[significant others: relative 1]

Participating in the group of significant others, the physiotherapist pointed out that restricted space not only hinders physical activity but also triggers a feeling of helplessness and entrapment.


*The resident is tied to the place.*
[significant others: relative 1]

The impression of being trapped in a certain area was more commonly perceived in buildings with multiple floors or in areas for residents with dementia. On floors with residents affected by dementia, there was usually no exit for residents for safety reasons. The only option for PA was to move within the restricted area. However, in multi-storey nursing homes, non-dementia residents were likewise affected by the impression of being trapped. In these cases, staircases were usually identified as the major obstacles. In principle, stairways were considered a good infrastructural element for physical training exercises. 


*Staircases are beneficial for PA, as [their use] results in the activation of more muscle groups (contracture prophylaxis), independence from the elevator, and the opportunity to move about the entire house.*
[significant others: physiotherapist 2]

However, staircases were not used or only rarely used as an opportunity to increase PA, not least because not all residents were physically capable to use stairs. 

Elevators allowed residents to move between levels and/or to leave the building. In this sense, elevators had a promoting effect for PA, although they are usually considered as impediments with regard to activity. However, to guarantee this effect, elevators have to work. In most nursing homes, this was not the case several times a year. 


*A defective elevator inhibits PA because movement is restricted to the living area under certain circumstances.*
[significant others: relative 2 and resident 1]

Most of the nursing homes included in our study have roof terraces and interior courtyards to offer residents the option of going for walks outside the building, regardless of whether they suffer from dementia or not. 


*In the interior courtyard residents with and without dementia [staff member 6] [residents] can walk around outside independently.*
[staff member 1]

Altogether, the photographs clearly document that designing nursing homes in a way that promotes PA is a complex challenge involving a wealth of day-to-day problems. This holds particularly true against the background that the primary goal of architectural planning is aimed at keeping residents safe, not at activating them. 

Accordingly, the photographs taken by staff and significant others often showed structures from the perspective of whether they were DIN standard compliant or not. According to the explanations provided for the photographs, they documented whether wide, obstacle-free corridors, functioning elevators, and outdoor terraces are available. In contrast, only a few photographs actually referred to residents’ PA or even depicted residents in motion.

#### 3.1.2. Opportunities and Limitations of Daily Work Equipment for PA Promotion

The theme ‘opportunities and limitations of daily work equipment for PA promotion’ (cf. examples Figure 2) summarizes equipment used in a resident’s daily routines with the potential to increase PA. The theme covers both the staff’s support equipment, such as lifters, as well as residents’ aids, such as rollators.

According to the staff, the residents’ first PA of the day is getting out of bed. The challenge here is that residents have to be able to fully hold their own body weight. 

If residents are unable to get out of bed, a lift is usually used for support. Staff members indicated that this relieves both the caregiver and the resident. However, the chance to train muscle strength through lifting their own body weight, and therefore, preventing progressing sarcopenia in the lower extremities, is minimized when using lifts.


*The device performs the task of getting up, not muscular strength.*
[significant others: relative 3]

Hence, using the lift in a way that promotes PA becomes particularly important. Comments by staff members emphasize that lifts offer more potential than merely helping residents out of bed, namely to also systematically train muscles in the upper extremities.


*[residents] need to use their arms and hands to grip and hold.*
[staff member 8]

The second option for physically activating residents is related to the choice of transfer during the day. There are three common types of transfer devices: rollator, walker, and wheelchair (with and without footrests). All of the supporting equipment provides opportunities for becoming physically active.

Staff members perceived walkers as especially beneficial for the residents’ PA, since their construction prevents falls.


*Walkers are beneficial as they reduce the risk of falling, are an alternative to manual fixation, rests are possible at any time.*
[significant others: physiotherapist 1]

Staff members also identified positive effects on residents’ PA from using rollators.


*Rollators are supportive because they provide a feeling of security, thus interrupting the fear-of-falling spiral, rests are possible, objects can be transported in the basket, thus allowing to focus on walking and not on carrying.*
[resident 3]

In the staff members’ opinion, wheelchairs also potentially foster PA.


*The wheelchair promotes the ability to move with one’s hands and feet, to overcome greater distances, rests are possible at any time.*
[staff member 10]

However, it was critically noted by staff that the fewer physical abilities a wheelchair-bound resident has, the more likely the resident tends to avoid PA. 


*The wheelchair is restrictive [for physical activity], because you let yourself be pushed, you become lazier and more comfortable.*
[significant others: former relative]

In order to solve this problem, a staff member suggested removing the wheelchair’s footrests, when possible. 


*Footrests inhibit PA because residents have no possibility to move by themselves and cannot move their feet and legs.*
[staff member 8]

The idea was to, thus, ‘force’ the respective persons to make a physical effort to move around. In this way, residents could perform ‘seated walking’ and pushing movements to move the wheelchair independently.


*Footrests promote PA in hemiparesis, leg prostheses, one leg on the footrest the other is moving.*
[staff member 8 and significant other: physiotherapist 2]

Taken as a whole, the equipment in nursing homes can be used to support, enhance, and challenge residents’ PA during the day. However, the use of equipment can also increase residents’ inactivity. Too much support can render equipment counter-productive. In this respect, the staff can be considered a decisive influencing factor as to whether residents are physically active or not, as they can decide to what extent equipment is used.

#### 3.1.3. Social Incentives for PA Promotion

The third theme ‘social incentives for PA promotion’ (cf. examples, Figure 3) is strongly informed by the residents’ perspectives and needs. It illustrates situations or circumstances in which PA functions as a supportive means to build or avoid social relationships. PA can facilitate social relationships that give residents a sense of responsibility and independence. Likewise, PA allows residents to escape the restrictions of the facilities’ daily routines.

One way to escape the daily routine is gardening. Staff members considered it helpful in preventing the deterioration of motor skills. From the residents’ perspective, maintaining motor skills is only one benefit of gardening; flower care or gardening also trigger a sense of responsibility, which in turn can lead to more engagement and concurrently more independent PA. 


*Flower care promotes PA, if flowers are not cared for, they will die and residents have to take care of them, no matter how they are doing.*
[resident 4]

The aspect of independent PA is also emphasized by significant others when explaining the positive effects of gardening.


*Familiar activities from the past (which had been conveyed/transferred to dementia), such as harvesting (e.g., of herbs or tomatoes), give a sense of success which then enhances the motivation to succeed.*
[significant other: relative 5]

Our study shows that residents can be motivated to be physically active especially through memories of past events. In this context, playing music seems to be particularly suitable for activating residents and helping them forget their physical limitations. 


*You can also move to music while sitting, you can’t move wrongly to music, it is often associated with positive memories/experiences, enhances motivation.*
[significant other: relative 6]

Pets can have a similarly motivating effect. Three of the participating nursing homes keep pets. Memories of owning pets in their former environments, but also the commitment necessary to build a relationship with the animal, can trigger activity in residents, e.g., motivate them to leave their rooms or the living area on a regular basis and go outside to feed a pet in the nursing home garden or to become active and cuddle a pet, such as a therapy dog, inside. All three stakeholder groups, residents, staff, and significant others took photographs of pets as an activity-enhancing factor while also documenting (via photographs of smiling faces) the enjoyment residents draw from interacting with pets. 


*Animals are allowed in the nursing home and are beneficial for the residents – provide many stimuli for PA.*
[resident 5]

Reference persons are also considered a distracting factor from physical limitations and pain. Some nursing homes employ medical clowns. The stakeholder groups documented this effect by taking photographs of medical clowns. According to staff members, the presence of the medical clowns not only distracts the residents but also leads to more PA because clowns initiated fun activities that would require residents to leave their rooms as well as leave their chairs during interaction and distract them from handicaps and limited mobility. 


*Escape from everyday life and distracts and [let them] forget their frailties.*
[staff member 1]

Staff, residents, and significant others attributed similar benefits to children. Children are considered a particularly valuable distraction from daily routines and one of the most effective activity-enhancing factors.


*Children encourage PA because they [residents] want to prove something, distraction by children, forgetting limitations.*
[resident 3]

Overall, encouraging independent action, creating opportunities to escape the homes’ daily routines, and encouraging residents to take on responsibility for people, beings, or things are the most effective strategies for physically activating residents in homes, according to the stakeholder groups.

### 3.2. Differences Regarding the Photographs Taken by Residents, Staff, and Visitors

Of all the groups residents had the most difficulty photographically documenting situations fostering activity. For this reason, they did not ascribe much validity to their photographs as documents of conditions that promote or hinder activity. The photographs’ motifs were rather considered representative of past experiences related to PA and high emotional value for the persons concerned (cf. examples between the groups Figure 4). The most important activity triggers are highly individualized, in that they are tied to the residents’ personal memories. In conversations with residents, it also became clear that the respective homes’ architecture plays only a subordinate role regarding their own PA, since it is perceived as the place one stays at and does not in itself have any activity-promoting effects. The staff, on the other hand, were attributed an important role in promoting PA, and it was repeatedly emphasized that staff members were doing everything they could to encourage residents to move. However, residents avoided taking photographs of staff during their daily work routine and did not document any infrastructural deficiencies impeding PA, which could indicate that they were primarily concerned with avoiding criticizing the nursing homes and those responsible.In contrast to the residents, the staff members considered their photographs representative of the activity-related situations in the respective nursing homes. From their perspective, the most important factors for keeping residents physically active were: (1) a safe, activity-friendly environment and (2) the employment of assistive equipment. Nursing home staff accordingly took photographs of all possible kinds of assistive equipment and its potential uses. This included not only equipment specifically for PA programs, such as balls or scarves, but also everyday devices, e.g., walkers or rollators. To capture the influence of the environment on PA, infrastructural conditions such as broken elevators, stairs, courtyards, or balconies were also photographed.Although several staff members ascribed a high activity-promoting potential to breaks or moving residents from one place to another, participating staff members likewise did not take photographs of other staff during breaks or when transferring an active resident in a wheelchair. Accordingly, daily care routines and associated mobilization exercises were not documented photographically.In the case of the significant others, the photographs mostly depicted activities in which they themselves or professionals were involved and in which they had a supporting role regarding the residents’ (physical) activity. In this respect, photographs documented events such as gait training executed by physical therapists or featured song books or pianos that were used when singing together.

Photographs taken by significant others either featured professional services (in the case of physiotherapists or priests) or documented infrastructural or architectural issues. No photographs were taken of staff performing mobilizing care. 

## 4. Discussion

This study deals with the factors that enhance or hinder residents’ PA in nursing homes. Photovoice was employed to this end from three different perspectives to capture the perceived factors influencing residents’ PA. From the staff’s perspective, the residents’ PA is determined by supportive aids and barrier-free environmental arrangements in the nursing homes. The significant others perceived their own influence on the residents’ PA as important. In addition, they attributed positive effects on residents’ PA to aids such as walkers or rollators. The greatest surprise to us was that the staff members’ and significant others’ perceptions were not reflected in the residents’ documentations. Residents also took photographs of assisting devices, such as walkers; however, they did not associate these devices with PA but rather considered them as means of transportation, e.g., for handbags or water bottles. From the residents’ perspective, a main incentive for becoming physically active was having responsibility for animals, flowers, or people (especially children). Activating triggers were, therefore, defined based on individual memories or relationships.

An aspect that needs to be considered for the discussion of findings is the particular context of the COVID-19 pandemic during which the Photovoice study took place. We conducted the concrete study in some of the homes prior to COVID-19-related restrictions and in some of the homes during times of restrictions. For most of 2020, participating nursing homes had to adapt their organizational routines and restrict group gatherings and activities, such as physical activity programs [49]. These adaptations might have influenced participants’ perceptions of options for PA and resulted in participants not including or documenting these opportunities in their photos but rather focusing on barriers or problematic situations.

However, in contrast to previous studies [50,51], the number of photographs featuring problematic and critical situations and constellations amounted to almost zero. This finding, as it was present particularly among residents, could be interpreted as avoidance behaviour, which was similarly identified in previous interview studies. Due to a feeling of dependence and the fear of negative consequences, the residents shy away from expressing criticism [38,46]. 

Our findings also confirm previous studies, which found that older adults ‘s perceptions of activity-promoting factors differ from other age groups’ [52]. PA itself has no relevance for the residents; it only becomes important when providing the opportunity to take responsibility or to socialize with people. These findings have considerable consequences for the design of PA promotion programs. Firstly, the resident needs to determine the subjective value of PA [39]. Secondly, activating situations have to be designed in such a way that they do not compromise the resident’s dignity and self-concept. This understanding, generated through the photographs, was already found to be helpful in previous studies [53]. 

What distinguishes our study from others is our explicit request that photographs be taken of situations and constellations serving or having the potential to serve the residents’ activity promotion. By enabling a change of perspective for all those involved, the photographs can, thus, reveal activation potentials in the nursing homes’ everyday life that would have been overlooked in a different approach. While earlier projects [38,50,54] conducted Photovoice studies with only one specific group of stakeholders, we decided to include perspectives of three different participant groups to examine potential differences in perceptions and consider the different groups’ agency for change. Similarly, to Wang and Burris [24], we aimed to integrate community members, i.e., residents, in the Photovoice study, while acknowledging that agency for community-level change and social action would rely on the hands of other participants groups, mainly key caregivers and staff in a nursing home setting. In nursing homes, daily routine and a specific form of lifestyle tends to be predefined by the institution and executed and supported by staff with very limited options for residents to structure their own daily activities [42].

As a consequence of this set-up, our study also illustrates the extent to which participant groups differed in their perception of activity-promoting and hindering situations and constellations in nursing homes. The key challenge in terms of resident activation, therefore, appears to be to create a critical consciousness among staff and caregivers for diverging perceptions and consideration of residents’ needs, to which the Photovoice project might contribute. Based on our findings, following staff’s perspective and providing aids or activation programs by professionals cannot be the only solution for physically activating residents. Rather, it might be key to have residents participate in daily activities and providing them with meaningful tasks and responsibilities. 

Such a shift in staff perceptions, then, would serve the ultimate purpose of any Freirean-based intervention: to co-create knowledge and cooperate for change and empowerment [55]. The photographs, here, could serve as a powerful visual tool to remind everyone to promote residents’ relationships with staff and significant others, encourage meetings with children and pets, as well as activities and connections to nature for nourishment.

### Limitations

One of the main limitations of this study can be found in the selection method for this sample. Mainly due to the COVID-19-related access restrictions, we had very limited options to meet and interact with significant others in the nursing homes. Thus, we were dependent on suggestions and recommendations by nursing home staff, which might have led to a selection bias, a general issue in Photovoice as identified elsewhere [33]. We tried to address this potential bias by aiming to ensure that every stakeholder group in the nursing homes had the opportunity to share their thoughts without any interruption or surveillance by others, e.g., by the ones who had suggested them for participation. At the same time, we aimed to create a context during the study so that every person living in the nursing home felt safe and appreciated. Special attention had to be paid to the circumstance that, in institutional settings such as nursing homes, everybody is acquainted, resulting in the risk that criticism is taken personally and leads to negative consequences [56,57]. This problem was also evident in our study, in that nursing home residents refrained from any criticism of infra- or staff-structural aspects.

Even though Photovoice represents a methodological tool that gives all participants space for sharing their specific perceptions, another limitation of our study is that we could only include residents who were physically and cognitively able to participate. We also cannot rule out that very inactive residents or residents feeling socially isolated were indirectly excluded. Furthermore, we cannot guarantee that the selected staff members or significant others were representative of the overall personnel, as they were included on a voluntary basis. 

Another limitation refers to the focus group discussions. Firstly, the executive staff of the nursing homes were given free hand in the composition of the focus groups. To prevent a potential bias effect caused by the selection of photographs used in the focus group, the compilation of photographs for the focus groups was organized by the first author of this study. Secondly, due to the on-going COVID-19 pandemic and access restrictions to nursing homes, we could not conduct the focus group discussions as planned but had to adapt the discussion format to a remote alternative developed by researcher DA. Thirdly, it was not possible for organizational reasons to include external collaborators or residents in the discussion.

The number of nursing homes included in the study was limited for research economics and organizational reasons. It would, therefore, be interesting for future studies to include a larger sample with a broader variety of residents. Future studies could also include other stakeholder groups, e.g., CEOs of umbrella organizations or politicians, to find out which relevance is attributed to physical activity promotion in nursing homes in general. In this regard, a larger Photovoice study with more stakeholder groups could create opportunities for dialogue on the importance of PA promotion in nursing homes.

## 5. Conclusions

Our results show how important it is for residents to be recognized and heard. Asking the participants to take photographs gave them the opportunity to think about reasons how and why residents become physically active. A surprising finding in this regard is that the most relevant trigger for residents to become physically active is the opportunity to do something independently or experience a connection with a person, a pet or an object prompting nostalgic feelings. This understanding could be useful for nursing home staff, i.e., to focus more on promoting biographical reflections when working with residents. Knowing positive emotional memories and life experiences related to PA can prove life-sustaining and help residents’ mobility. To the best of our knowledge, the potential benefits of biography work as a means of PA promotion in old age has not been thoroughly researched yet. 

Looking back on our observations, we have come to the opinion that it is impossible to develop a nursing home concept promoting PA in a way that appeals to everyone to the same extent. Nevertheless, it is imperative to note that the expectations of professionals regarding activity triggers and barriers may differ significantly from aspects that trigger PA in residents. For this reason, an ongoing dialogue between staff and residents is an essential prerequisite for successful PA promotion programs. Because in the end, according to our overall impression, it is the dialogue between the stakeholder groups that above all guarantees the sustainability of healthy behaviours in the residents by offering them the chance to decide by themselves what to do with the time given to them.

## Figures and Tables

**Figure 1 ijerph-18-05481-f001:**
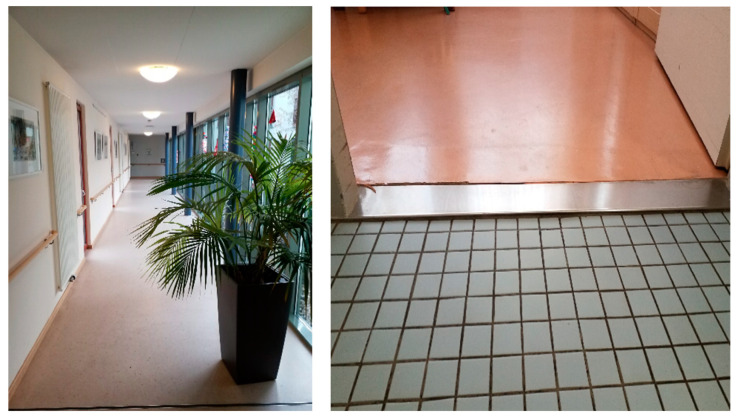
Examples of photographs on the implementation of norm standards vs. not standard conforming building.

**Figure 2 ijerph-18-05481-f002:**
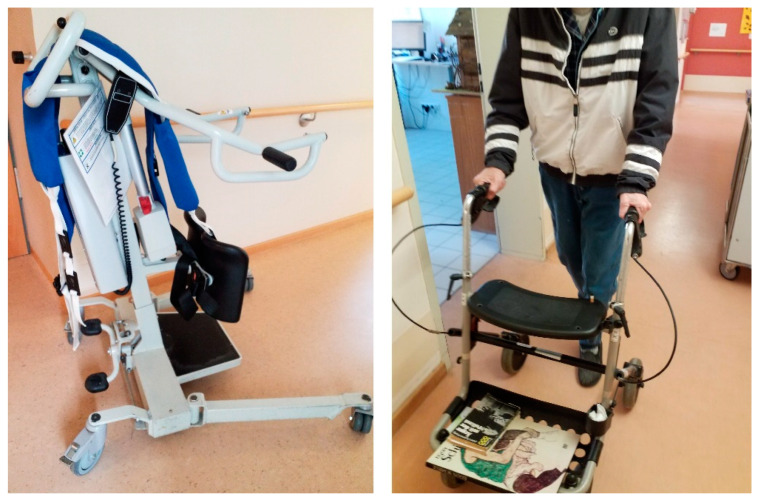
Examples of photographs by staff of how to enhance PA vs. photographs by residents of how enhance their PA.

**Figure 3 ijerph-18-05481-f003:**
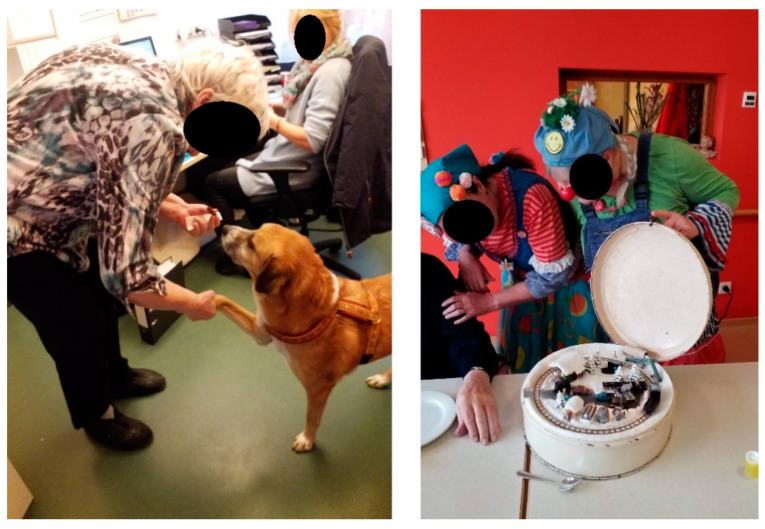
Examples of photographs with social interaction types.

**Figure 4 ijerph-18-05481-f004:**
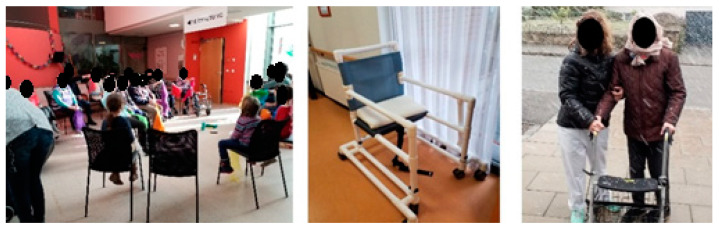
Differences in photographs taken, 1. residents, 2. staff, 3. significant others.

**Table 1 ijerph-18-05481-t001:** Overview of number of photographs taken per home and by the different perspectives.

Nursing Home	No. of Photographs	Residents	No. of Photographs	Staff	No. of Photographs	Significant Others	Number of Photographs
1	13	0	0	2	8	1	5
2	24	0	0	1	4	2	20
3	9	1	1	1	8	0	0
4	42	2	4	1	9	2	29
5	9	0	0	1	5	1	4
6	28	1	6	1	17	1	5
7	40	2	8	2	30	1	2
8	4	0	0	1	4	0	0
Total numbers	169	6	19	10	85	8	65

## Data Availability

The data presented in this study are available in depersonalized form on reasonable request from the corresponding author. The data are not publicly available due to confidentiality reasons and to allow participants anonymity.

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
