# Peer review of "‘All We Have to Decide Is What to Do with the Time That Is Given to Us’ a Photovoice Study on Physical Activity in Nursing Homes†"

_ijerph, 2021, doi:10.3390/ijerph18105481_

Round 1
Reviewer 1 Report
General comment (originality, scientific accuracy, strengths and/or weaknesses):
The objective of the study was clear and it has been well led by the introduction. The numerous limits of the study, highlighted by the authors in the part discussion, demonstrated that the current results must be taken with caution but these results are interesting. One would expect a larger number of people to be included (n = 6) due to the large number of nursing homes (n = 8). The results of the study show a trend that will be confirmed in the future.
Minor comments:
Line 210: what is green dot or red dot?
Insert in the main text the numbers of the Figure
Reviewer 2 Report
Overall the authors present an current issue and used a very creative way to answer their questions. The introduction provides sufficient background but can be improved. For a transparent insight the methods need to be described more detailed. For more information please see the attached document.

Reviewer 3 Report
It was a pleasure to review this paper. The entitled paper “All we have to decide is what to do with the time that is given to us*” – A Photovoice Study on Physical Activity in Nursing Homes by Dorothee Altmeier, Ansgar Thiel and Annika Frahsa aims to explore, through qualitative methodology, the important factors in NH that influence residents PA behaviour and to compare the perceptions of the different participant groups (residents, significant others and staff).
This manuscript is well written and well-structured and addresses a very pertinent topic. Indeed, it urges a better understanding of the factors that promote or inhibit PA in NH residents. However, I would suggest that the authors address the following minor comments:
1) The abbreviation NH presented twice in the abstract (line 10 & 11), it's rarely used during the manuscript. The authors should adopt the same nomenclature throughout the manuscript.
2) Line 137, the authors refer to the BaSAlt Study (supplementary). Could you clarify this “(supplementary)” meaning? It was supposed to be a supplementary file with information about the BaSAlt Study?
3) Line 180/1, the authors state that “Most significant others were suggested and informed by the nursing 180 homes’ executive staff and recruited by a research team member”. There were any criteria for this recruitment? Could this way of recruitment (being suggested) lead to a bias of the results?
4) “The participants were asked to take photographs of situations, objects…” (Line 183). This instruction included a minimum or a maximum number of photos that each participant should take?
5) Several NH had to adapt and change internal circuits and several activities were suspended due to the COVID-19 pandemic (e.g. residents could leave the floor where their room belong or usual activities group activities suspended). These changes could influence residents PA opportunities. Therefore, maybe it will be interesting to understand if during this study the selected NH had a temporary restriction in the spaces or activities that would normally be available to the residents.
